# Unique Features and Collateral Immune Effects of mRNA-LNP COVID-19 Vaccines: Plausible Mechanisms of Adverse Events and Complications

**DOI:** 10.3390/pharmaceutics17101327

**Published:** 2025-10-13

**Authors:** János Szebeni

**Affiliations:** Nanomedicine Research and Education Center, Department of Translational Medicine, Semmelweis University, 1085 Budapest, Hungary; jszebeni2@gmail.com

**Keywords:** LNP, lipid nanoparticle, mRNA, Comirnaty, Spikevax, COVID-19 vaccinations, adverse effects, gene therapy, immunotherapy, pandemics

## Abstract

A reassessment of the risk-benefit balance of the two lipid nanoparticle (LNP)-based vaccines, Pfizer’s Comirnaty and Moderna’s Spikevax, is currently underway. While the FDA has approved updated products, their administration is recommended only for individuals aged 65 years or older and for those aged 6 months or older who have at least one underlying medical condition associated with an increased risk of severe COVID-19. Among other factors, this change in guidelines reflect an expanded spectrum and increased incidence of adverse events (AEs) and complications relative to other vaccines. Although severe AEs are relatively rare (occurring in <0.5%) in vaccinated individuals, the sheer scale of global vaccination has resulted in millions of vaccine injuries, rendering post-vaccination syndrome (PVS) both clinically significant and scientifically intriguing. Nevertheless, the cellular and molecular mechanisms of these AEs are poorly understood. To better understand the phenomenon and to identify research needs, this review aims to highlight some theoretically plausible connections between the manifestations of PVS and some unique structural properties of mRNA-LNPs. The latter include (i) ribosomal synthesis of the antigenic spike protein (SP) without natural control over mRNA translation, diversifying antigen processing and presentation; (ii) stabilization of the mRNA by multiple chemical modification, abnormally increasing translation efficiency and frameshift mutation risk; (iii) encoding for SP, a protein with multiple toxic effects; (iv) promotion of innate immune activation and mRNA transfection in off-target tissues by the LNP, leading to systemic inflammation with autoimmune phenomena; (v) short post-reconstitution stability of vaccine nanoparticles contributing to whole-body distribution and mRNA transfection; (vi) immune reactivity and immunogenicity of PEG on the LNP surface increasing the risk of complement activation with LNP disintegration and anaphylaxis; (vii) GC enrichment and double proline modifications stabilize SP mRNA and prefusion SP, respectively; and (viii) contaminations with plasmid DNA and other organic and inorganic elements entailing toxicity with cancer risk. The collateral immune anomalies considered are innate immune activation, T-cell- and antibody-mediated cytotoxicities, dissemination of pseudo virus-like hybrid exosomes, somatic hypermutation, insertion mutagenesis, frameshift mutation, and reverse transcription. Lessons from mRNA-LNP vaccine-associated AEs may guide strategies for the prediction, prevention, and treatment of AEs, while informing the design of safer next-generation mRNA vaccines and therapeutics.

## 1. Introduction

Since the roll-out of mRNA-based COVID-19 vaccines in December 2020, Pfizer’s Comirnaty and Moderna’s Spikevax, over 5 billion doses of these vaccines have been distributed globally, corresponding to an estimated 2–3 billion individuals vaccinated with one or more doses [1]. The success of these vaccines in combatting COVID-19 has accelerated the adoption of lipid nanoparticle-formulated mRNA (mRNA-LNP) technology for developing innovative vaccines and therapeutic agents targeting infectious diseases, metabolic disorders, cancer, and various other conditions. However, like all other medical interventions, the mRNA vaccines also face challenges, such as the emergence of an unusually broad spectrum of severe adverse events (AEs) and long-term complications, collectively referred to as “post-vaccination syndrome” (PVS) [2,3,4,5,6,7].

Appendix A lists the typical AEs of PVS, classified by affected organ systems. The compilation demonstrates the uniquely broad spectrum of symptoms impacting nearly all organ systems. Importantly, several of these AEs also overlap with typical manifestations of COVID-19 and post-COVID syndrome, which have been designated by the Brighton Collaboration, an international team of vaccine experts, as being “AEs of special interest” (AESIs) [8,9,10,11,12]. Although these AESIs are rare (affecting approximately 0.1–0.4% of recipients [13], the millions of vaccine injuries resulting from the massive scale of global vaccination with mRNA vaccines has led to increasing concern about the risk–benefit ratio of the procedure in healthy people once the life-threatening pandemic subsided. Highlighting the scale of interest, a compilation of 767 experimental and clinical studies on PVS was recently published [14]. The concern has recently led to a transition in the USA from universal mRNA vaccine use to risk-based administration, as detailed in the Outlook.

Given the atypical and wide-ranging nature of mRNA vaccine-induced AEs, traditional interpretations prove inadequate, suggesting that we may be dealing with a previously unrecognized disease entity. In fact, the purported pathogen of this new disease, i.e., the mRNA-LNP, is a protein corona-free streamlined analog of SARS-CoV-2 virions wherein the viral envelope is replaced by a polyethylene-glycol-decorated phospholipid layer, and the viral core is represented by a monocistronic mRNA bound to synthetic aminolipids assumably via hydrogen bonding [15]. Given that all three basic vaccine elements may be toxic in susceptible individuals, those afflicted with PVS might have been exposed to cumulative toxic risks.

One paradoxical difficulty in this field is the excess of descriptive data on individual AEs, with little attention given to the fundamental questions about the root cause(s) of toxicity, potentially stemming from the mRNA-LNP structure. Hence, the goal of this perspective review is to assemble and explain the potential contributing factors underlying various AEs, and, for the first time, link them to some unique structural features of these vaccines. Highlighting the inherent limitations of the mRNA-LNP technology may offer valuable lessons to guide the R&D of future mRNA-LNP-based vaccines and pharmaceuticals.

## 2. Essentials of Natural Immunogenicity and the Mode of Action of Conventional Vaccines Versus mRNA-LNP

To understand the originality of mRNA-LNP-based vaccines, it is helpful to revisit how the immune system naturally processes and presents antigens upon exposure to exogenous pathogens such as SARS-CoV-2, and how traditional immunization works.

In a nutshell, natural pathogens are taken up by professional antigen-presenting cells (APCs) such as dendritic cells, macrophages, and B cells through phagocytosis, receptor-mediated endocytosis (e.g., via Fc receptors), or pinocytosis at the site of infection. Inside APCs, the pathogen is enclosed in phagosomes, which fuse with lysosomes to form phagolysosomes. Within this compartment, the pathogen is degraded by proteases into peptides, which are then presented in narrow grooves on the surface of MHC (Major Histocompatibility Complex) class II molecules (in short: class II) exposed on the surface of APCs. These peptide-MHC class II complexes, displayed on APC surfaces, are recognized by CD4^+^ helper T cells and B lymphocytes, initiating an adaptive immune response [16,17,18]. This pathway of immunogenicity involves uptake, structural modification, and transport of antigenic peptides through the endoplasmic reticulum (ER) and Golgi apparatus. Each step is finely regulated by external and internal control factors that modulate proteolytic enzyme activity, peptide selection, processing, trafficking, and stability of the antigen-class II complexes. Such control factors include proinflammatory cytokines, Toll-like receptor agonists, invariant chains (Ii/CD74), HLA-DM and HLA-DO chaperones, and inflammasomes. Their combined activity establishes the complex immune-stimulatory state known as inflammation.

Traditional immunization follows the above scheme, except that denatured or synthetic, nonpathogenic but immunogenic components, mainly proteins or peptides derived from pathogens, are taken up by APCs near the injection site and/or draining lymph nodes. The subsequent immune stimulation required for adaptive defense is generated by adjuvant ingredients in most vaccines, such as aluminum salts, oil-in-water emulsions, Toll-like receptor agonists, or saponin-based adjuvants, with alum being the most widely used adjuvant worldwide.

In sharp contrast, in mRNA vaccines the only immunogenic component, the spike protein (SP) is pathogenic itself, and it is generated all over the body, not only in APCs but also in many other cell types that take up the vaccine nanoparticles. Furthermore, these vaccines do not contain a classical adjuvant; rather, the LNP together with the mRNA provides the adjuvant effect. Such deviation from natural infection- and vaccine-induced adaptive immune response represents an unprecedented new approach to inducing immunity against pathogens. The process is considered by many as infection-preventive immune therapy or immuno-gene therapy [13], rather than conventional immunization or vaccination.

## 3. The Rise, Impact, and Reassessment of the mRNA-LNP Vaccine Platform

The evolution of nucleic acid transfection methods spans nearly four decades, beginning with lipofectin used as a delivery vehicle [19], followed by cationic liposomes [20], and more recently LNPs [21,22,23,24,25,26,27,28,29].

A key milestone in the medical application of mRNA-LNPs was the introduction of nucleoside modifications [30,31,32,33,34,35,36], which reduced the proinflammatory effects of mRNA and enhanced its translational efficiency. These advances culminated in the deployment and official recommendation of mRNA-LNP vaccines as “effective and safe” during the COVID-19 pandemic and beyond, leading to the vaccination of billions worldwide, until their recent revocation in the USA for the general healthy population, with recommendations now restricted to certain high-risk groups, as detailed in Section 6.

While recognizing the scientific breakthroughs brought by mRNA vaccine technology contributing to all these achievements, it is crucial to reiterate the above-described fundamental divergence from traditional vaccination, warranting thorough scrutiny of its biological consequences. Unlike most presently used vaccines that act by reproducing the tightly regulated process of natural immunogenicity, the direct delivery of genetic instructions for the SP bypasses key regulatory checkpoints in the natural pathway of antigen processing and presentation, thereby overriding mechanisms that evolved over 500 million years [37,38,39]. This deviation from nature carries the potential risk of incontrollable, sustained and dysregulated immunogenicity, leading to atypical, adverse immune responses in certain individuals.

## 4. Distinct Structural Features of mRNA-LNP Vaccines Contributing to Collateral Immune Effects and Adverse Phenomena

Table 1 compiles a list of structural and functional features of Comirnaty and Spikevax that may, in theory, be associated with certain AEs reported for these vaccines. They represent intrinsic features of these formulations, distinct from other vaccine types, but with conceptual similarity to DNA-based “genetic” vaccines. The two vaccines cause similar PVS, although the intensity of individual symptoms may vary. The unique properties are listed in the 1st column of Table 1 and linked to experimentally and/or clinically demonstrated adverse immune processes in column 2. These abnormal immune reactions are then associated with different AEs in column 3 based on cell biology and immune pathology extrapolations.

For clarity, it should be emphasized that Table 1 does not suggest that all listed processes contribute to AEs, nor that these are the only mechanisms. It intends to be a compilation of currently recognized cellular and molecular mechanisms that theoretically may explain the variety of symptoms associated with LNP-mRNA PVS. It is certain that the COVID-like vaccine AEs rising in a few out of hundreds of vaccine recipients are multicausal; each can arise from one or more of the listed or other adversarial pathways, acting independently or simultaneously, in an additive or synergistic fashion. The rise and extent of AEs depend on many individually varying genetic and epigenetic factors, as well as external and internal conditions.

### 4.1. Ribosomal Translation May Alter Antigen Fate and Function

As delineated in Section 2, a critical step in the immunization with currently used traditional vaccines is the presentation of small (12–26 amino acid, AA) immunogenic peptides to T helper (Th) and B lymphocytes in narrow recesses (grooves) on the surface of class II molecules [17]. However, the SP synthesized on ribosomes is nearly 100 times larger than the peptides typically presented by class II molecules as it consists of 1273 AAs, more than twice the size of hemoglobin (574 AA) or albumin (585 AA). This large protein is synthesized in the cytosol outside of the phagolysosomal compartment, so the cell recognizes it either as an aberrant cytosolic protein or as an intracellular pathogen, directing it into the intracellular antigen-processing pathway where the protein is digested by the proteasome into short peptides (8–11 AA) that are loaded onto MHC class I molecules in the endoplasmic reticulum (ER) for subsequent presentation to CD8^+^ cytotoxic T cells [40,41]. Such deviation from the evolutionarily optimized pathway of antigen processing and presentation could result in uncontrolled cytoplasmic accumulation of the protein with unpredictably diversified downstream consequences.

#### 4.1.1. Diversification of SP Processing and Presentation

Figure 1 illustrates that the vaccine-induced immunogenicity of the SP via ribosomal synthesis bypasses the evolutionarily conserved and tightly regulated pathway of external antigen processing and presentation. This may entail numerous abnormal immune processes, including proteasome digestion of the SP that may lead to autoimmunity-triggering cross-presentation of SP peptides; SP secretion that can entail multiorgan toxicity; expression of the SP on cell surfaces that may trigger autoimmune attacks and antigen-independent polyclonal T cell activation with systemic inflammation. Additional risks include exosomal excretion of the SP that may contribute to systemic toxicity and multiorgan transfection with a toxin, while excessive somatic hypermutation and SP frameshift mutations potentially cause autoimmunity and other unprecedented pathologies. Finally, reverse transcription of the mRNA may result in insertion mutagenesis with protein aberrations and cancer. These collateral immune effects are discussed in detail in the subsequent subsections.

Figure 2 breaks down the unconventional pathways of SP processing and presentation, serving as a visual guide for their detailed discussion in the following subsections.

#### 4.1.2. Proteasome Digestion, Resulting in Cross-Presentation of SP Peptides on Class I Molecules Triggering Autoimmunity

Proteasomes are multi-subunit protein complexes in the cytosol that hydrolyze ubiquitin-tagged, unneeded, misfolded, or foreign proteins. Proteolysis occurs in the barrel-shaped central part of proteasomes, while a regulatory cap unfolds and translocates the antigenic peptides into the ER of dendritic cells and other APCs. There they are loaded onto MHC class I molecules and subsequently presented on the cell surface to CD8^+^ cytotoxic T lymphocytes (Tc cells). As illustrated in Figure 2, this can be the fate of the SP, too, although as a foreign antigen is should primarily be presented to CD4^+^ helper T cells via class II molecules [42,43]. As a consequence, the SP peptides can trigger cytotoxic Tc attacks; one way to induce autoimmune phenomena that are associated with mRNA-LNP vaccinations [43,44,45,46]. However, the cross-presentation of SP peptides on class I molecules does not necessarily lead to autoimmune attack against the APC by CD8^+^ Tc cells. It is known that if these T cells express high-affinity receptors for self-peptides presented on MHC class I, they undergo apoptosis in the infant thymus, a process known as clonal deletion, thereby preventing autoimmune phenomena. Normally, Tc-s attack only cells that display foreign peptides on class I molecules and simultaneous co-stimulatory signaling also occurs.

Figure 3 outlines details of this multistep synergistic activation process; the receptor–ligand surface proteins and cytokines secreted by the two cell types. In the absence of co-stimulatory signaling, the Tc cells may become anergic or tolerized and may protect the foreign antigen-presenting APCs against cytotoxic attack [42,43,44,45,46,47]. Based on this, it can be hypothesized that autoimmune attack against APCs occurs when the mRNA vaccines’ strong proinflammatory effect (discussed later) provides the costimulatory signals to the Tc-APC interaction. Such duality in CD8^+^ T cell actions may explain the alternative fates of SP-expressing transfected cells: either pathogenicity or Tc-induced tolerance, explaining the long-term persistence of the SP in blood and affected tissues.

#### 4.1.3. Spike Protein Secretion

The beige-colored area in Figure 2 illustrates a second potential anomaly in SP processing: its secretion into the circulation. The secreted SPs are synthesized by ribosomes bound to the rough ER (RER), which constitute the smaller fraction of the total ribosome population in human cells [48]. The SPs then enter the ER lumen for folding and initial modification. From there, they are transported via vesicles to the Golgi apparatus, where further processing and sorting occur. Finally, the proteins are packed into secretory vesicles that fuse with the plasma membrane, releasing their contents into the extracellular space through exocytosis.

Spike protein secretion into the extracellular space may have two outcomes: (1) receptor-mediated reuptake by APC and then standard processing and presentation, responsible for conventional immunogenicity, and (2) distribution into the circulation, causing multiorgan toxicities [14,49]. The former phagocytic reuptake and processing through the standard phagolysosome-ER-Golgi-class II pathway plays an essential role in the anti-SP immunogenicity of the vaccine, which is beneficial and accounts for antiviral protection. However, the SP that escapes into the circulation may lead to toxemia, as the SP can act as a facultative toxin and thus represents a major inherent cause of vaccine-induced AEs. The presence of free SP in the blood and its extended stay in the circulation will be addressed later in detail.

#### 4.1.4. Spike Protein Expression on Cell Surfaces

While viral capsid proteins act as immunogenic shells, the SP expressed on cell surfaces exerts long-lasting systemic adverse effects, hence, it represents a toxic gain-of-function. The phenomenon is analogous to the assembly of viral SP on the plasma membrane prior to the budding of intracellularly replicated viruses from infected cells. This pathway follows the conventional secretion route, except that from the trans-Golgi network, SP-containing vesicles are directed to the plasma membrane, where they fuse with the phospholipid bilayer exposing the S1 subunit with its receptor-binding domain (RBD) [50,51,52]. This decoration, literally “coronation” of cell membrane patches may lead to two types of antibody-mediated cell damage and to antigen-independent polyclonal T cell activation, as detailed below.

##### 4.1.4.1. Antibody-Dependent Cellular Cytotoxicity

One example of the antibody-mediated damage of SP-exposing cells is an innate cellular response known as antibody-dependent cellular cytotoxicity (ADCC). It is mediated by NK cells, macrophages, and neutrophil granulocytes (Figure 4A), whose Fc receptors bind to the Fc portion of SP-bound antibodies and release apoptotic effector molecules, like those released by CD8^+^ T cells upon binding to MHC class I molecules.

##### 4.1.4.2. Complement Activation

The other, antibody-mediated damage of cells expressing the SP is due to complement activation, which can proceed via all three standard routes: the classical, the alternative, and the lectin pathways. These are illustrated in Figure 4B, from top to bottom, respectively. The classical pathway is activated by the binding of SP-specific IgG and IgM to SP on the cell surface, among them, the neutralizing antibodies which serve the anti-virus function of the vaccine. These antibodies bind to C1q, triggering the autocatalytic cascade that culminates in the formation of the membrane attack complex (MAC), which perforates the lipid bilayer of activating cells.

As for activation via the alternative pathway, a recent in vitro study showed that this affects peripheral blood immune cells (PBMCs), inducing the secretion of IL-1β and TNF-α [53]. Since inhibition of complement activation significantly reduced cytokine release, the finding suggests a direct or indirect causal relationship [53]. Finally, the mannose-rich SP on cell surfaces can also activate complement via the lectin pathway, by binding mannose-binding lectin (MBL) or ficolin. The result in all three pathways is the same: MAC-mediated cytotoxicity.

The role of complement activation in vaccine-induced AEs is further discussed in the context of PEGs used for vaccine stabilization.

##### 4.1.4.3. Antigen-Independent Polyclonal T Cell Activation Due to Superantigen Activity of the SP

Yet another potential consequence of SP exposure on immune and other cells is the triggering of antigen-independent polyclonal T cell activation. This rare immune phenomenon is caused by bacterial superantigens, for example, *Staphylococcal Enterotoxin B* and *Streptococcal* pyrogenic exotoxins, which are extremely potent immune activators that can cause toxic shock syndrome in infected children [54,55,56,57]. These toxins include a 10–20 amino acid sequence, termed superantigen motif, that crosslinks APCs and T cells by simultaneous binding to the extra-groove region of MHC class II (and/or CD28) molecules on APCs and to the Vβ region of the TCR (Figure 5A).

The concept that the SP can act as superantigen was based on the sequential similarity between the bacterial superantigen motif and SARS-CoV-2 residues T678–Q690 in the S1 subunit. Importantly, the rare, highly lethal multisystem inflammatory syndrome (MIS) observed in mRNA vaccine recipients has identical symptoms as the toxic shock syndrome [54,56,58,59,60,61,62,63,64,65,66,67], suggesting similar pathomechanism [55,56,57,68,69]. Notably, the superantigen motifs, not found in other coronaviruses [59], can fundamentally upset the immune system by stimulate up to ~20% of T cells, although conventional antigens activate only ~0.0001–0.001% of the body’s T-cell pool [70]. As for the molecular-cellular interactions involved in such striking toxicity, the superantigen motif is located near the site where furin cleaves the SP into S1 and S2 [71,72,73]. It is also close to a strongly positive (+3) short amino acid sequence, known as the PRRAR motif, which can bind the SP to negatively charged (sialylated) membrane domains containing, for example, heparin proteoglycan. Thus, there is a unique molecular constellation whereupon the PRRAR tightens the superantigen motif to cell membranes [74,75,76] and flags the site for furin cleavage, entailing S2-mediated membrane attachments or fusion [56,59,60,61,62,63,64,65,66,67]. The SP1 shedding, on the other hand, may amplify the toxic consequences of cytokine release. Overall, this mechanism mirrors natural SARS-CoV-2 infection, where viral binding to ACE2 is followed by furin-mediated cleavage of SP into toxic S1 and fusogenic S2, the latter enabling viral entry.

Just as with staphylococcal superantigens, one prerequisite for nonspecific APC-T cell binding is the presence in the circulation of the superantigen motif-containing SP subunit, specifically S1. In the case of mRNA vaccines, this was initially not assumed to occur, as the original concept held that the entire SP would remain intracellular, serving solely as an antigen for adaptive immune education. However, as mentioned earlier (Section 4.1.3), free SP can appear in the bloodstream and persist for months [77,78,79,80].

A further complication in implicating the SP in superantigen-like action arises from the observation that, unlike bacterial superantigens, the SP alone did not exhibit intrinsic superantigen-like inflammatory activity [81]. These findings led to the hypothetical model illustrated in Figure 5C, proposing that membrane expression of S1 on the surfaces of APCs creates the unique molecular configuration required for nonspecific T-cell engagement with robust immune activation.

Clarifying the contribution of superantigen-like interactions to a small subset of the most severe vaccine-related injuries is of great importance, as it could open avenues for pharmacological intervention. Currently available examples include heparin [82], which neutralizes the polybasic insert, and a monoclonal antibody 6D3 [83], which can inactivate the superantigen motif [83].

Beyond this APC-T cell bridging, further nonspecific cell aggregation caused by the superantigenic motif on SP may contribute to pathology. These include red blood cell aggregation (rouleaux formation) [84,85,86], erythrocyte–platelet binding [87,88,89,90,91], and interactions with non-ACE2 receptors, such as CD147 on lymphocytes and other immune cells [92,93,94,95,96,97].

#### 4.1.5. Exosomal Excretion of the Spike Protein

An additional abnormal processing pathway of the SP is via excretion of exosomes by APC and transfected body cells (Figure 2D). Exosomes are small extracellular nanovesicles (30–150 nm) that play a role in intercellular communication and carry biomolecules such as lipids, proteins, and nucleic acids, including mRNA. In the case of mRNA vaccine-transfected cells, they can carry the SP on their surface, as well as the whole mRNA and its immune stimulatory small fragments called miRNAs (microRNAs) [98,99,100]. Exosomes propagate from cells in a manner similar to how newly formed virus particles emerge from SARS-CoV-2-infected cells, key differences being that the nucleocapsid is replaced with SP-crowned plasma membrane and the mRNA does not code all virus proteins. Since the lipids from LNPs may incorporate into SP-expressing exosomes, these vesicles may represent virus/LNP chimeras. They can contribute to the immunogenicity of the SP by presenting antigens to APCs just as the vaccines do, providing rationale to use exosomes as an alternative to LNP-based vaccines [101,102,103]. The demonstrated efficacy of this approach for inducing immunogenicity attests to the exosomes’ capability for transfecting distant tissues and cells and thus contributing to the inflammatory and/or autoimmune complications. Experimental confirmation for these AEs was provided by Bansal et al. [100] who demonstrated that circulating exosomes containing SP appear in the bloodstream shortly after administration of the BNT162b2 vaccine, even prior to the development of detectable neutralizing antibodies. This finding established exosomes as an early mediator of immune activation and a plausible contributor to the systemic distribution of vaccine-derived antigens [99]. Consistent with these observations, Giannotta et al. [104] reviewed molecular pathways whereby spike-laden exosomes can promote vascular and cardiac inflammation through endothelial uptake, TLR4 signaling, and NF-κB-driven cytokine release, thereby offering a mechanistic explanation for myocarditis and other inflammatory adverse events [104].

Beyond inflammatory activation, exosomal transport of SP has also been implicated in the perturbation of homeostatic systems. Bellavite et al. [105] highlighted how spike protein, carried systemically by exosomes, may interfere with the renin–angiotensin system, disrupt endothelial integrity, and propagate hyperinflammatory cascades. Taken together, these studies suggest that exosomal dissemination functions as a “Trojan horse” mechanism, spreading spike protein from locally transfected cells to distant tissues. This pathway provides a biologically plausible link between mRNA vaccination, spike protein biodistribution, and the occurrence of multisystem adverse events.

Because mRNA-SP-carrying exosomes and SARS-CoV-2 share some basic features, exosome-recipient cells can be considered as “pseudo-infected” with the virus. In essence, the mRNA-LNP undergoes virtual metamorphosis from sterile nanoparticles into dumbed-down, quasi-infectious pseudo-viruses, capable of spreading transfection similarly to live attenuated viruses. Such virus-mimetic “pseudo-superinfection” may contribute to the AEs associated with mRNA vaccines [99].

#### 4.1.6. Excessive Somatic Hypermutation in B Cells

Somatic hypermutation (SHM) is a phenomenon whereby the immunoglobulin genes encoding the variable region of the B-cell receptors (BCRs) undergo an especially high rate of mutation to produce slightly different versions of the antibodies after the B cells are activated though interaction with APCs and helper T cells. The B cells that produce high-affinity antibodies against the antigens survive and undergo affinity maturation, while those with lower-affinity antibodies die off. The process gives rise to plasma cells, which secrete antibodies that are more effective at neutralizing the target antigen, as well as the production of long-lived memory B cells, which ensure quick production of high-affinity antibodies at a later encounter with the pathogen. Thus, SHM enhances the effectiveness of the antibody response, helping the immune system better combat the pathogen targeted by a vaccine.

The mRNA vaccines are potent inducers of follicular helper and germinal center B cell responses in the spleen and lymph nodes, driving further acceleration of somatic hypermutation and affinity maturation beyond physiological levels [106]. The robust SHM can persist for at least 3 weeks and contributes to the superior antibody response to mRNA vaccines. However, if tolerance checkpoints fail, SHM also carries a risk of producing antibodies with cross-reactivity with host cell antigens, generating autoreactive clones [107] that contribute to post-vaccination autoimmune manifestations [108]. Taken together, these findings suggest that an overdrive of SHM elicited by mRNA vaccines, while highly effective at promoting protective immunity, may simultaneously elevate the risk of autoantibody generation and autoimmune pathology [109,110,111,112,113,114].

#### 4.1.7. Polymorphism Due to Frameshift Mutation

Another adverse consequence of the potential buildup of stabilized SP in the cytoplasm of APCs and other transfected cells is frameshift mutation, i.e., an error during the translation of chemically modified mRNA due to a slippage of codon identification by transfer RNAs. This viral mechanism for immune escape via surface variation may lead to SP polymorphism, resulting in the formation of undefined peptide products and paraproteins with unknown antigenic and autoimmune potential.

The experimental evidence for frameshift mutation includes a study by Mulroney et al. [115] who have shown the formation of two additional SP bands over the in-frame expected product. In another study, Boros et al. found that vaccination with 1mψ-mRNA can elicit cellular immunity to peptide antigens produced by +1 ribosomal frameshifting in major histocompatibility complex-diverse people [79]. The translation of 1-mψ-mRNA was shown by liquid chromatography-tandem mass spectrometry to contain 6 in-frame and 9 chimeric SP peptides [79]. The complications of frameshift mutation include autoimmunity, neurotoxicity, and the weakening of humoral immunity [116,117,118]. The link between frameshift mutation and Tc-mediated autoimmunity was shown by the significantly increased IFN-γ response to frameshifted antigens which was observed only in individuals vaccinated with Comirnaty [79]. Another adverse consequence is a highly significant increase in heart muscle 18-flourodeoxyglucose uptake, which was detected only in vaccinated patients up to half a year [79]. Guanine-quadruplex (G-quadruplex) formation, which makes the RNA secondary structures particularly stable and is implicated in post-vaccination neurotoxicity [119], is also thought to be a consequence of frameshift mutation.

#### 4.1.8. Reverse Transcription of the mRNA with Insertion Mutagenesis

The accumulation of chemically stabilized SP mRNA in the cytoplasm of APCs, or any other mRNA-LNP-transfected cells, increases the risk of reverse transcription, whereby the mRNA is copied into complementary DNA (cDNA) by reverse transcriptase enzymes. The resulting free-floating DNA in the nucleus or cytoplasm, called episomes, can be integrated into the genome by other enzymes in a process known as insertional mutagenesis. Several mechanisms can mediate insertional mutagenesis, involving transposons, integrases, and DNA repair enzymes, such as topoisomerases. Transposons possess endonuclease activity, allowing them to move within the double helix and induce various sequence alterations, including the insertion of reverse-transcribed mRNA fragments. This mechanism, carried out by “LINE-1 retrotransposons” has been implicated in the integration of reverse-transcribed SARS-CoV-2 DNA into cultured human cells, which was subsequently re-transcribed into viral mRNA, providing a plausible explanation for persistently positive PCR tests in some long-COVID patients [120,121]. Such “neo-gene” formation may occur if the reverse-transcribed sequence remains intact and is coupled with promoter activity, potentially leading to permanent genomic alteration. In the case of the SP, this could theoretically result in chronic autoimmunity or toxicity. Additionally, topoisomerase-mediated error repair may integrate plasmid sequences during the unwinding and re-ligation of DNA.

Because insertional mutagenesis is largely random, neo-gene formation is not the only possible outcome. Reverse transcription-mediated integration may also disrupt essential genes or regulatory elements, leading to loss of function or disease, such as cancer, if tumor suppressor genes are inactivated or proto-oncogenes are activated.

A newly described potential pathway of mRNA-driven insertional mutagenesis involves human DNA polymerase theta (Polθ, EC 2.7.7.7), a low-fidelity polymerase in mammalian cells mainly engaged in RNA-templated DNA repair [122,123]. Studies demonstrated that Polθ can use RNA templates to synthesize cDNA and promote its integration into DNA [124], thus, if spike cDNA or its fragments accumulate in the nucleus, Polθ could insert them into the genome, at least theoretically [125,126]. However, other studies have suggested that reverse-transcribed viral cDNA is detectable in only a small fraction of cells, and genomic integration is even rarer, particularly in the context of vaccine-derived mRNA [127]. Among others, Merchant et al. also challenged the notion of spike gene integration into chromosomal DNA following mRNA vaccination [128].

Considering these uncertainties, the question remains open: what explains the detection of the SP in blood for up to 187 days [129], or up to 180 days in cardiac and skeletal muscle cells at sites of inflammation and fibrosis [130], when the longest-circulating proteins in the blood, IgG, have a half-life of 21–28 days? In fact, numerous studies have reported the prolonged presence of full-length SP in the blood or other organs [80,129,131,132,133,134,135,136,137,138,139,140]. These observations are difficult to reconcile solely with the increased stability of the mRNA and/or the SP.

In summary, while the mechanistic possibility exists for the incorporation of the SP genetic code into the human genome, the actual frequency and biological significance of such events remain uncertain.

### 4.2. The Spike Protein Can Cause Systemic Toxicity

#### 4.2.1. The Structure and Toxicities of the Spike Protein

The SP is a trimer glycoprotein that forms the well-known spikes (peplomers) of the SARS-CoV-2 virus. It is essential for the binding of virions to ACE-2 and the later identified CD147 receptors, facilitating subsequent fusion and the release of viral mRNA into the host cell. However, unlike most viral surface extensions, the SP appears to fundamentally impact essential cellular functions, a unique property that has been the subject of intense research, as evidenced by >2600 records on Medline (National Library of Medicine) when searching (by EndNote in August 2025) for the combined “All Fields” search terms “spike protein”, “COVID-19”, and “mRNA vaccine”. Therefore, its secretion by APCs, or by any other transfected cells, implies toxicosis or toxoidemia with sporadic toxicity.

Table 2 compiles a list of potential toxicities of the SP, free or in exosomes, underlying the pluripotent pan-toxicity of the protein. These include oxidative mitochondrial damage [36,99,141,142], the upregulation of proinflammatory cytokines [143,144] and other destructive processes in cells that produce SP after vaccine uptake by target and/or non-target cells [80,145,146,147,148].

As for the collateral damage in vaccine non-target cells, mounting evidence points to endothelial cells as primary targets of SP toxicity. In vitro studies with human ECs have shown that the S1 subunit enhances inflammatory cytokine release and triggers NF-κB activation through ACE2 and/or C3a receptor engagement [145,148,160]. Furthermore, S1 stimulates the generation of exosomes, reflecting endothelial injury [160]. Collectively, these data substantiate the role of circulating SP in driving endothelitis [161] and consequent multisystem inflammation syndrome [162].

#### 4.2.2. Vaccinology Perspectives of Spike Protein Toxicity

Because vaccination with Comirnaty or Spikevax inevitably leads to the secretion of toxic SP, yet clinical toxicity occurs only sporadically and overlaps with that caused by live SARS-CoV-2, these vaccines resemble attenuated live-virus or toxoid-based vaccines, which only rarely induce symptoms of the very disease they are designed to prevent. A historical example of vaccine pathogenicity is paralytic poliomyelitis, where the weakened poliovirus in the oral formulation regains virulence and causes polio symptoms [163]. Likewise, the diphtheria and tetanus toxoids in the DTaP vaccines can occasionally revert to being toxic [164,165,166,167,168], causing the symptoms of infection. Thus, although the mechanisms are entirely different, the mRNA vaccines pose a risk of acting like SARS-CoV-2, i.e., it may be viewed as a facultative pathogen. Support for the latter concept was provided recently, when Moderna’s Phase 1 clinical trial of its investigational RSV vaccines (mRNA-1345 and mRNA-1365) was paused in July 2024 after several severe lower respiratory tract infections were reported in RSV-naïve infants [169,170].

### 4.3. Multiple Chemical Modifications of the mRNA and SP Increase Their Function and Stability

Beyond the above modifications of antigen synthesis, processing and presentation, a profound deviation from natural immunogenicity is achieved by multiple chemical modification of the mRNAs. Their combined effect is a substantial increase in the biological stability and translation activity of the mRNA relative to native mRNAs. The specific modifications are discussed below.

#### 4.3.1. Replacement of mRNA Uridine with Pseudouridine (ψ)

Extracellular unmodified mRNAs have strong immune stimulatory effect due to their binding by pattern recognition receptors, such as Toll-like receptor-3 (TLR3), TLR7, and TLR8, or the retinoic acid-inducible gene I (RIG-I) receptor [40]. This effect, along with the instability, poor translation efficacy, and high cost of mRNA synthesis prevented therapeutic use of mRNAs prior to the 2010s. It was therefore a milestone in mRNA research that Kariko et al. replaced uridine with N(1)-pseudouridine (N1-ψ) in the mRNA nucleotide chain, to reduce its immune reactivity and increase its biological stability and translation efficacy [30]. The transfection of mice in these experiments with pseudourinated luciferase mRNA using lipofectin as an mRNA carrier resulted in a 78-fold increase in splenic luciferase translation 24 h after the injection, showing the feasibility of lasting mRNA transfection and translation using a complex of ψ-mRNA with a fusogenic lipid [31]. Subsequent studies [33,34] further improved the efficacy of gene translation, particularly with N1-methyl-pseudouridine (m^1^ψ), wherein a methyl group is attached to the N1 position of the uracil ring. This uridine analog outperformed ψ in protein expression [171].

#### 4.3.2. Codon Optimization

Codon optimization implies changing bases in the polynucleotide sequence in the mRNA to match the human codon for amino acid tRNA, thus improving the stability and translation efficacy of the SP. This molecular method ensures high levels of SP expression in human cells for robust immune response, while maintaining the stability and functionality of the mRNA [172].

#### 4.3.3. Methylation of the 5′ Cap

Methylation of the 5′ cap in mRNA vaccines refers to a chemical modification of the mRNA molecule at the 5′ end. The addition of a methyl group improves mRNA stability by resisting exonucleases and enhances the recognition of the mRNA by the host cell’s ribosomes, leading to efficient translation into the target protein, i.e., SP. The 5′ Cap methylation also minimizes unwanted immune responses against the synthetic mRNA, thus improving vaccine efficacy [173,174,175,176].

#### 4.3.4. UTR Stabilization

UTR stabilization involves modifying the untranslated regions (UTRs) of mRNA to enhance its stability and translational efficiency in vaccines. Thus, Comirnaty utilizes a 5′ UTR derived from the human alpha-globin gene and a 3′ UTR combined from the amino-terminal enhancer of split and mitochondrial ribosomal RNA 1 genes. Spikevax adopted a 3′ UTR from globines. The stability and efficient translation of the mRNA enhance the production of the target antigen, resulting in a stronger immune response [98,177,178].

#### 4.3.5. Poly(A) Tail Optimization

The 3′ poly(A) tail optimization in mRNA vaccines refers to the engineering of the polyadenylated tail at the 3′ end of the mRNA to enhance its stability, efficiency of translation, and overall efficacy in SP expression [179]. The poly(A) tail containing 100–150 adenine residues protect the mRNA from degradation by exonucleases in the cytoplasm, thereby extending its half-life. It can also interact with the 5′ cap of the mRNA to form a closed-loop structure, facilitating efficient ribosome binding and protein synthesis. The tail also helps in the nuclear export of mRNA and proper localization in the cytoplasm [98,177,178].

#### 4.3.6. GC Enrichment of the mRNA

The enrichment of the SP mRNA with guanine (G) and cytosine (C) bases (GC enrichment) aims to reduce innate immune stimulation of the mRNA. This is because uridine-rich sequences strongly activate innate immune receptors (TLR7/8, RIG-I, MDA5), whereas GC-rich sequences lower uridine frequency and stabilize the mRNA secondary structure, making the molecule less recognizable to RNA sensors [180,181,182,183]. Accordingly, GC enrichment of the SP mRNA reduces proinflammatory cytokine induction [30,184,185]. Furthermore, by allowing the formation of stable secondary mRNA structures (hairpins or stem-loops) around translation initiation sites, GC enrichment has an impact on ribosome access, translation initiation, codon usage and elongation speed. GC enrichment of mRNA also promotes G-quadruplex (G4) formation, leading to altered protein yield and folding kinetics. The overall consequence can therefore be either the promotion or hindrance of the extent and diversity of SP synthesis [186,187,188,189].

#### 4.3.7. Proline Enrichment of the Spike Protein

Although the common understanding originally promoted was that the SP mRNA and recombinant SP would persist in the body only for a few days or weeks, later experimental and clinical studies revealed that modified SARS-CoV-2 mRNA and the recombinant protein can persist in the body for months after the injection. Among many reports, Boros et al. detected SP mRNA in the cardiac and skeletal muscle at sites of inflammation and fibrosis up to a month, while the SP was shown to persist in the blood a little over half a year [79]. Ota et al. found full length SP in the cerebral arteries 17 months post-vaccination [190].

One of the explanations for the persistence of the SP is the double proline substitution, also called the “2P mutation” [191,192]. The process was introduced into the SARS-CoV-2 spike to stabilize the protein in its prefusion conformation [191,193,194]. Proline has a distinctive cyclic structure where the amino group is connected to the side chain, making it a “secondary” amine rather than the usual “primary” amine found in other amino acids. This can cause kinks or bends in protein structures, which influences protein folding, stability, and flexibility, reducing its metabolism and prolonging its circulation time. In addition, the 2P substitution limits S1 dissociation from S2, also known as S1 shedding, and favors the intact trimer, which may contribute to extended antigen persistence [195].

Furthermore, proline and hydroxyproline emerge as prominent deuterium (heavy hydrogen) binding sites in structural proteins lending robust isotopic stability to the protein that resists not only enzymatic breakdown, but virtually all (non)-enzymatic cleavage mechanisms known in chemistry [196]. This property too can contribute to the SP’s widespread distribution throughout the body and toxicities.

### 4.4. The LNP Is a Strong Stimulant of Innate Immune Responses, Also Enabling mRNA Transfection

The use of LNPs as mRNA carriers was an essential milestone in the development of genetic vaccines [23,29]. This is because, at least in part, in addition to protecting and carrying the mRNA, the LNPs are strong stimulators of the innate immune system and thus serve as adjuvants to the vaccine that enables its potent immunogenicity. This adjuvant effect seems to be particularly effective, since the LNPs cause many-fold lymphocyte and plasma cell proliferation in the germinal centers of lymph node follicles, where the antibodies are formed [106,197,198]. Although the exact mechanism of this remarkable immune stimulation is not clear, one likely contributor is the fact that the LNPs can simultaneously trigger the activation of both the cellular and humoral arms of the innate immune systems, specifically, leukocytes and the complement (C) system. The two activations can amplify each other in a positive feedback loop.

#### 4.4.1. Activation of the Cellular Arm of Innate Immunity

The mRNA-LNPs are strong activators of innate cellular immunity. In Comirnaty, the primary contributing factor is the ionizable aminolipid ALC-0315, which accounts for 46% of the lipid content [199]. Experiments on mice and monkeys have shown that the LNP can bind to Toll-like receptors of immune cells, most importantly TLR2 and TLR4, as well as to other danger signal receptors. This leads to the NF-κB pathway of Th1-dominant proinflammatory cytokine and chemokine secretion, including IL-1β, IL-2, IL-6, IL-18, IFN-γ, TNF-α, and GM-CSF [200,201,202]. These LNPs also cause NLRP3 inflammasome activation that entails caspase-1-mediated apoptosis [202]. The stimulation of cellular immunity is further enhanced by the anaphylatoxins C3a and C5a [203,204], products of the concurrent stimulation of the C system by mRNA-LNPs

#### 4.4.2. Triggering of Humoral Immune Response: Complement Activation

The fact that liposomes and many other nanoparticulate drugs and agents can activate the C system has been known for decades [205,206,207,208], but for mRNA-LNPs it was shown only in a 2022 pig study [209]. This study investigated the mechanism of the allergic and anaphylactic reactions to Comirnaty that occurred shortly after the start of vaccinations in December 2020 [199,210]. The symptoms of these reactions were very similar to the infusion reactions caused by intravenous injection of liposomes, which were shown earlier to be related to C activation [211]. The study by Dezsi et al. provided evidence that Comirnaty can cause C activation-related pseudoallergy (CARPA) [209], suggesting that the human vaccine reactions could be attributed, at least in part, to CARPA [199]. This theory was supported later by Barth et al. [212] and Barta et al. [213].

A further adverse consequence of C activation by the LNPs is accelerated disintegration in the lymph and blood due to the formation of the membrane attack complex (MAC, SC5b-9) resulting in membrane damage with premature release of mRNA payload [214,215,216,217]. Due to the above adverse impacts of anti-PEG antibodies on PEGylated drug products, the FDA recommends screening for both anti-protein and anti-PEG antibodies [218].

As mentioned regarding the mechanism of C activation by mRNA-LNPs (Section 4.1.4.2), an in vitro study revealed that it proceeds via the alternative pathway [53]. It was also shown in PBMCs that C activation may play a causal role in the vaccine-induced secretion of IL-1β and TNF-α [53]. In addition to the alternative pathway activation, the LNP exposure to high levels of anti-PEG antibodies in plasma leads to classical pathway activation. Further pig experiments showed that such activation can cause anaphylactic shock [213], explaining the increased incidence of anaphylaxis after mRNA vaccination [209]. However, unlike the constitutive C action by the LNP lipids, the substantial inter- and intraindividual variation in anti-PEG antibody levels in the blood makes this type of antibody-dependent C activation sporadic. It should also be noted that the SP is also a C activator via the lectin pathway [147,149]. Thus, at different stages after vaccinations, different C activation pathways can get involved, putting the body under constant inflammatory pressure.

#### 4.4.3. The LNP Is a Superadjuvant

Adjuvants are vaccine ingredients that enhance the antigen-specific immune responses. The mRNA vaccines do not contain any added adjuvant, yet they are very efficient inducers of anti-SP response. The above delineated simultaneous activation of the cellular and humoral innate immunity by the LNPs, which may be additive and/or synergistic, may explain the potent adjuvant effect of LNPs. Among the experimental findings demonstrating superior innate immune stimulation by LNPs, one of the most widely publicized observations is the robust proliferation of follicular helper T cells, germinal center B cells, B memory cells, and plasma cells in the lymph nodes of mice following LNP immunization, accompanied by the release of proinflammatory cytokines and chemokines [106,219,220,221]. Hence, in the context of mRNA-LNP vaccines, the LNP may be considered as a pharmacologically active ingredient [222], a “super adjuvant”.

#### 4.4.4. The LNP Is a Fusogenic Transfecting Agent

Another major adverse impact of ionizable lipids is their fusogenic activity. Since the original mission of the type of LNP used for vaccinations was the transfection of cells in gene therapy, the selection of ionizable lipid component was based, among others, on its fusogenic potential. Actually, the clinical success of liver-targeted patisiran (Onpattro), the first FDA-approved gene therapy against amyloidosis, which used a fusogenic lipid in the LNP [25], was an important contributor to the selection of ALC-0315 for Comirnaty, although the latter was not intended to directly target the liver or other organs. As shown in Figure 6, the ionizable fusogenic lipids used in Onpattro, Comirnaty, and DOPE/DOSPA, components of the classical transfection agent, lipofectamine [22], are very similar in having an ionizable amino group at one end, and multiple membrane-affine fatty acid chains, at the other. It is therefore not surprising that Comirnaty transfects cells beyond the APCs in the lymph nodes draining the deltoid muscle.

The capability of LNPs to deliver mRNA in different body cells was shown, among others, by Pardi et al. in 2015 [223]. In their study, intense fluorescence emerged in the liver 24 min after deep muscle injection of luciferase mRNA-LNP in mice [223]. It was also observed that superficial muscle injection entailed less protein translation in the liver, suggesting that the injection site and depth are critical variables in LNP spreading [223]. More relevant to the vaccine campaign, in a preclinical study by Pfizer/BioNTech, a tritiated lipid marker was used to explore the biodistribution of Comirnaty-equivalent luciferase-mRNA-LNP in rats [200]. The study showed 2.8% of radioactivity in the plasma 15 min after the LNP injection, peaking between 1 and 4 h and distribution mainly into the liver, adrenal glands, spleen and ovaries in >2% over 48 h [200]. However, <2% radioactivity was also seen in 12 other organs. Yet in further studies, the Ψ-mRNA was detected in the brain, heart, liver, spleen, ovaries, testes, and bone marrow and blood after vaccination of rats with mRNA-LNPs. Filling the gap in LNP tissue distribution studies in large animals, Ferraresso et al. [224] found that exogenous protein expression occurred in all major organs in swine when injected i.v. with a relatively low dose of mRNA encapsulated in a clinically relevant LNP formulation. Exogenous protein was detected in the liver, spleen, lung, heart, uterus, colon, stomach, kidney, small intestine, and brain, as well as in circulating white blood cells and platelets, and bone marrow megakaryocytes and hematopoietic stem cells. These results showed that nearly all major organs “are viable targets for mRNA therapies” [224], while the coins’ other side is that it may be difficult, if possible at all, to exclusively target certain cells with LNPs without off-target effects.

These and other data, reviewed by Pateev [28], attest to rapid multiorgan distribution and transfection of body cells with the SP mRNA. The consequence is the same as described for APC above, proteasome processing, and Tc-attack that can lead to multiorgan autoimmune phenomena.

### 4.5. The PEG on the LNP Surface Is Immune Reactive and Immunogenic

While the intrinsic C activation by the vaccine lipids is likely to be a key contributor to the LNPs’ superadjuvant effect, mRNA-LNPs can activate C via another way, too, depending on an extrinsic factor: access to anti-PEG antibodies. This activation proceeds via the classical pathway and may synergize with the constitutive activation by the lipids via the C3 amplification loop [225]. The main consequences of such occasional C activation are discussed below.

#### 4.5.1. True and Pseudoallergic Reactogenicity

The first alarming AEs, observed shortly after the vaccination campaign began in December 2020, was a cumulation of allergic reactions to the vaccine, some escalating to anaphylaxis [210]. Similar reactions were reported in many other vaccination centers in different countries, leading to new guidelines for vaccine eligibility, excluding people with severe allergies. Nevertheless, the incidence of anaphylaxis following mRNA vaccination has been markedly higher than that observed with other vaccines, approximately ~60-fold increase compared with flu vaccines [13].

Since these reactions cannot be linked to specific allergy against any vaccine ingredient, the most likely mechanism is C activation-related pseudoallergy (CARPA) [209,226]. Solid evidence for a causal role of anti-PEG antibodies in C activation-related anaphylactic shock was recently obtained in pigs [213].

#### 4.5.2. Anti-PEG Immunogenicity

Considering the bridging role of C activation between innate and adaptive immunity [227,228], C activation by the LNPs is also likely to play a causal or co-stimulatory role in the anti-PEG immunogenicity of mRNA vaccines. Such activity was recently shown by Kozma et al., in recipients of Comirnaty and Spikevax, after the second and third booster injection [229]. As shown in Figure 7A,B, the levels of anti-PEG IgG and IgM rose over the pre-vaccination baseline with an order of magnitude in the case of Spikevax, while Comirnaty caused a smaller but significant rise in anti-PEG IgM after the second booster injection (Figure 7B). The greater immunogenicity of Spikevax compared to Comirnaty is consistent with the results of Carreno et al. and Ju et al. [230,231], and the finding of Comirnaty’s immunogenicity based on a significant rise of anti-PEG IgM after the second dose (Figure 7B) was confirmed by Bavli et al. for anti-PEG IgG [232], showing its significant, >2-fold increases three weeks after immunization with Comirnaty [232].

A direct correlation between blood anti-PEG antibody levels and allergic reactions was demonstrated in the significantly higher anti-PEG IgG and IgM in AE reactors compared to non-reactors (Figure 7C,D) [229], implying increased risk for anaphylaxis in people with high anti-PEG antibody levels. In the same study, Kozma et al. [229] also revealed the presence of preexisting anti-PEG IgG and IgM in 98–99% of 118 healthy unvaccinated blood donors, among whom 3–4% were “anti-PEG Ab supercarriers” displaying extremely high antibody levels (Figure 7E,F) [229]. The right-skewed histograms, characterized by many low values and a few very high ones, together with the log-normal distribution of antibody levels (insets in Figure 7E,F), in which variability increases with the mean, are typical of multicausal, synergistic biological processes [40].

### 4.6. The LNP Is Unstable in Water

The nanoparticles in mRNA-LNP vaccines are not as stable as the traditional therapeutic liposomes. At the pH of human blood or tissue fluids, the ionic interaction between the ionizable lipid (BNT-0315) and the negatively charged mRNA might lose dominance, and the complex may be held together, at least in part, by hydrogen bonds [15]. Also, as revealed by electron microscopic images, the phospholipid bilayer coat of Comirnaty nanoparticles lacks continuity, and mono- and bilayers alter, implying less packaging and barrier functions compared to liposomes. A part of the LNPs is bicompartmental with mRNA-free blebs budding out, which was associated with increased transfection potency [233].

Nano-mechanically, the Comirnaty NPs are soft, compliant structures, that were shown by multiple experimental approaches to be in a “liquid state” [234]. These unique features of mRNA-LNPs may contribute to the inherent instability of vaccine NPs whose shelf-life, after opening the thawed vials, is 24 h [235]. An electron microscopic study showed that during long storage at 4 °C, the Comirnaty NPs are prone to disintegration, yielding snake-like winding nano-segments hypothetically identified as mRNA lipoplexes [15]. This is illustrated in Figure 8, which shows an electron microscopy image of Comirnaty LNPs stained with uranyl acetate after storage at 4 °C. While other therapeutic liposomes show no morphological changes during such storage time, most of the nanoparticles in Comirnaty display major transformations, mainly fusion and disintegration, with the appearance of worm- or snake-like structures, presumably mRNA-lipoplexes.

### 4.7. The Injectable Vaccines May Contain Contaminations with Plasmid DNA and Inorganic Elements

Several studies reported different undeclared contaminants in the Pfizer mRNA vaccine including segments of the plasmid DNA vector used as the template for in vitro transcription of SP DNA to mRNA and double-stranded mRNAs [236,237,238]. As discussed in Section 4.1.8, integration of DNA fragments into the host DNA can be shown in vitro, but the evidence for their contribution to “turbo” malignancy in vivo is heavily argued in the cancer field.

DNA contamination in the vaccine as a source of cancerogenic mutagenesis was demonstrated, among others, by Kammerer et al., who found large amounts of residual DNA in Comirnaty that included the monkey Simian virus’ cancerogenic promoter/enhancer sequence, known as SV40 [239]. As for other contaminants, Diblasi et al. [238] reported the presence of 55 undeclared chemical elements in six different brands of COVID-19 vaccines, including Comirnaty and Spikevax [238]. Except for noble gases, the undeclared elements were from all groups of the periodic table, some with luminescent and magnetic properties. Beyond the cytotoxicity of lanthanides, high concentrations of the heavy metals detected have been associated with toxic effects in humans. The range of detected elements showed substantial variation across different COVID-19 vaccine brands and sampling times, for example, in Comirnaty, >20 different elements were detected in different batch samples at different times, with only a few declared in the vaccine’s specification. This inconsistency led the authors to the conclusion that the differences arise because of the dynamic nature of the nanoscale self-assembling nanostructures over time. In fact, other studies report different exotic sub- or multimicron particles in the mRNA vaccines, such as ribbons, sheets, nanotubes, nano dots, and nano scrolls [240]. These represent lipid self-assemblies, mRNA lipoplexes or different ion-lipid complexes and crystals, graphene oxide, calcium carbonate with graphene inclusions, iron oxide, PEG, and many more multi-molecular complexes.

Nonetheless, neither the source of these contaminations nor their contribution to AEs was determined and remain to be explored in the future. The fact that they were present in all vaccines points to a systemic issue with the physical and chemical properties of the vaccines, rather than to a fault of any production process or batch.

In the lack of experimental data or hypotheses regarding the causality between molecular self-assemblies and AEs, one theoretical mechanism is C activation. The causal role of this phenomenon in AEs was discussed in the context of adverse LNP effects (Section 6.4), which also applies to alternative nanoarchitecture including vaccine contaminations. As is known, urate, hydroxyapatite and cholesterol crystals [241,242,243,244,245], organic and inorganic phosphates [246,247], needle-structures [206,248] or other artificial, non-biological surfaces can activate the C system involving both the classical and the alternative pathways. Complement activation, as mentioned earlier, represents a major proinflammatory stimulus.

## 5. Turbo Cancer

The close temporal relationship of haemato-lymphoproliferative disorders with mRNA vaccinations, mainly after booster injections, led to the concept of “turbo cancer” [249,250,251,252,253,254], a highly debated assertion on the acceleration of malignancy and rise in rapidly progressing new cancers following vaccinations with mRNA vaccines. Beyond leukemias and lymphomas, there are case reports on sarcomas, glioblastomas, keratoacanthomas, myocardial myxoma, and bladder tumor, temporally associated with vaccinations with or without genetic evidence. One of the hypothetical explanations is insertion mutagenesis, i.e., the insertion of cancerogenic DNA sequences into the host DNA, such as the mentioned (Section 4.7) monkey Simian virus’ cancerogenic promoter/enhancer sequence, SV40 [239].

Among the experimental and clinical data supporting the turbo cancer concept, Nacionales et al. described ectopic lymphoid tissue proliferation in mice injected with mRNA [255], whose features are reproduced by malignant B and T cell lymphomas and lymphoid leukemias, whose frequency has substantially increased over the past few years [253,256]. Gentilini et al. reviewed 28 reports on malignancies developed soon after mRNA-LNP COVID vaccination, and 26 of them were B-cell and T-cell lymphoproliferative disorders [256]. The authors argued that the time correlation between vaccination and symptoms met the Hill’s criteria for causation, a group of nine principles establishing epidemiologic evidence of a causal relationship between a presumed cause and an observed effect [257].

In addition to the presence of SV40 tumor promoter in E-coli DNA plasmid contamination of mRNA vaccines [239], further explanations for turbo cancer include (i) immune exhaustion (lymphopenia) with suppression of type I interferon response [98,258], (ii) inhibition of p53 SP tumor suppressor protein [259], (iii) tumorigenic effect of methylated pseudouridine [260,261,262], (iv) the blocking of innate and specific IgG1 antitumor responses by nonspecific IgG4 through inhibitory Fc receptors (called fragment crystallizable gamma receptor IIb, FcγRIIB) on B cells, dendritic cells, and macrophages [263]. Furthermore, (v) small, non-coding micro-RNAs (miRNAs, ~22 nucleotides) in exosomes and (vi) G-quadruplexes (four-stranded mRNA formed in guanine-rich regions) in modified RNAs have also been associated with vaccine-induced tumors, among others, via downregulating tumor suppressor genes (e.g., miR-21) [119,250,264,265,266].

Despite all these studies and theories, critics of the turbo cancer concept keep challenging the credibility of temporal relationships and/or in vitro correlations as evidence for a causal relationship between vaccination and cancer. This concept is not accepted by mainstream medicine, for example Wikipedia considers it as an anti-vaccination conspiracy theory (September 2025). Along this line, there are studies showing the anticancer effect of mRNA vaccines, e.g., Ramos da Silva et al. provided evidence that the mRNA vaccines prevented tumor relapses, eradicated subcutaneous tumors at different growth stages, and suppressed human papilloma virus-caused cervical cancer in mice [249].

## 6. Outlook

### 6.1. The Pendulum Swing of mRNA Vaccines

It is common view that mRNA-LNP-based vaccines have played a pivotal role in combating COVID-19 under emergency use authorization, although the number of saved lives is under debate [267,268,269,270,271,272]. The technology is considered as a revolutionary advancement in vaccine science [273,274,275], offering significant advantages such as simplified, accelerated, and cost-effective production. Its flexibility allows for rapid adaptation to viral mutations and the potential to deliver multiple antigens simultaneously, enabling the development of combined vaccines against multiple viral strains. The assertions by the manufacturers, regulatory agencies (WHO, CDC, FDA), major healthcare institutions, and other authorities that these vaccines were effective and safe enabled continued promotion and unforeseen investment into this technology. In 2024, the US FDA approved Moderna’s second mRNA vaccine for RSV (mRESVIA, mRNA-1345) [276,277] and later the Omicron variant XBB.1.5-adapted and KP.2-adapted formulations [278]. Most recently, in late August 2025, the FDA approved Moderna’s NEXSPIKE (mRNA-1283) and Pfizer’s Comirnaty, LP.8.1-adapted monovalent formulations.

Beyond the fully authorized COVID-19 and RSV vaccines, >50 Phase I-III clinical trials were in progress against COVID-19, infectious diseases and cancer [178,279,280,281], and over 300 mRNA-LNP-based drugs were in preclinical development by scores of companies. The mRNA-LNP-based antiviral vaccine trials were against influenza, cytomegalovirus, Epstein–Barr virus, varicella-zoster virus, herpes simplex virus, Zika virus, norovirus (stomach flu), and HIV [178,279,280,281]. The global market for mRNA vaccines and therapeutics was projected to be in the multibillion-dollar range and was predicted to grow several-fold within a few years [282,283,284]. Clearly, the mRNA-LNPs technology was thought to have remarkable promise for personalized, more effective treatments for a range of unmet medical needs.

Since the end of the pandemic, the use of all COVID-19 vaccines has naturally declined, including the mRNA-based ones. As of mid-2025, the mRNA vaccine pendulum began to swing in the opposite direction as several notable changes in CDC and FDA policies reflect a clear shift from universal to risk-based recommendations of mRNA vaccines [285,286]. Most recently (from late August 2025), the FDA, CDC, and HHS have jointly restricted the recommendation of all mRNA vaccines only to adults ≥ 65 years and individuals with underlying high-risk conditions, such as asthma, obesity, diabetes, immunosuppression, and those with chronic illnesses [287,288,289]. These events led to highly polarized public discourse around vaccinations with mRNA and other vaccines, underscoring the need to reinvigorate vaccine research in the near future.

### 6.2. Novel Medical Perspectives from the mRNA Vaccine Era

Among the unforeseen outcomes of the mRNA vaccination campaign is the emergence of PVS, a relatively rare iatrogenic orphan disease entity not observed prior to the use of mRNA-LNP vaccines [13]. Unlike classical vaccine adverse reactions (local pain, fever, mild systemic symptoms), PVS is characterized by a broad range of persistent, multi-organ, and often severe manifestations (Appendix A), whose clinical profile markedly differs from most known post-vaccination AE syndromes. The phenomenon challenges the traditional vaccine safety-monitoring framework and calls for a paradigm shift in pharmacovigilance; one that embraces long-term, systems-level surveillance of emerging vaccination and other immunogenetic therapies.

Another new medical perspective is APC-targeted autoimmunity, representing an immune attack against the very cells that “teach” the immune system for adaptive response. In a metaphorical sense, the phenomenon mirrors the fate of Seneca, compelled by his student Nero to commit suicide. Importantly, the “Seneca phenomenon” may contribute to the dysregulation of immune responses that may underlie some AEs addressed in this review.

### 6.3. A Unifying Message Amidst Abundant Data and Conclusions

Post-vaccine syndrome is evidently multicausal, whose full understanding awaits a comprehensive integration of the diverse anomalous mechanisms involved. With this goal in mind, the central concept emphasized in this review is that most AEs can be traced back to inherent structural features of the vaccine, pointed to by the spikes of a schematic SARS-CoV-2 virus in Figure 9.

### 6.4. Recommendations for the Future

To mitigate the recurrence of PVS-like phenomena in the future, several measures may be considered. These include the implementation of modernized safety surveillance systems and evidence-based guidelines, the broadening of autopsy investigations to capture a wider spectrum of pathologies, the establishment of more stringent product qualification criteria, and enhanced pharmaceutical accountability with transparent risk communication [290]. Beyond these pragmatic steps, elucidating the underlying mechanisms and developing targeted strategies for the prevention of AEs remain of substantial importance for improving the safety of future products based on the mRNA-LNP technology [290]. With many milestones already achieved, and the momentum reached, an upward swing in this field is inevitable, carrying the medical utilization of mRNAs into a new phase of growth, innovation, and lasting impact.

## Figures and Tables

**Figure 1 pharmaceutics-17-01327-f001:**
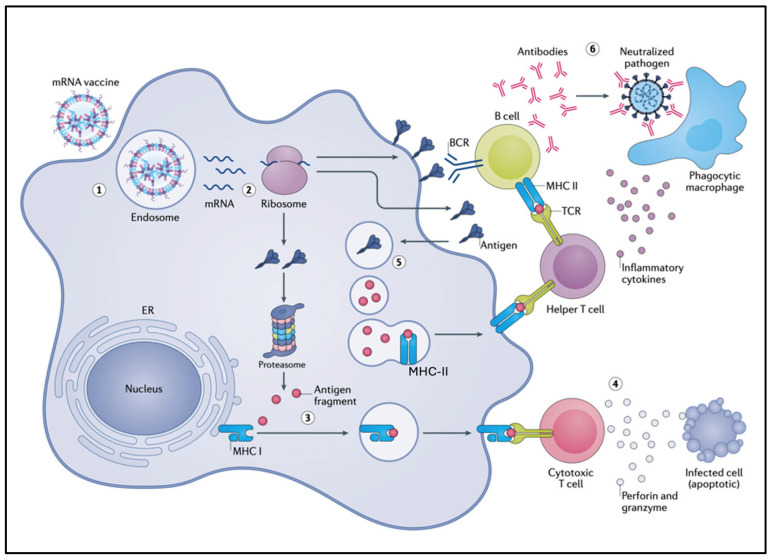
The mechanism of mRNA-LNP vaccine action. The figure, reproduced from Ref. [40], illustrates the steps via which the vaccine induces immunogenicity against the SP in dendritic or other APCs. The uptake of mRNA-LNPs via phagocytosis (step 1) is followed by the release of the mRNA from the endosomes and translation on ribosomes to the full-length SP trimer (step 2). Thereafter the SP may be expressed on the cell membrane as seen on SARS-2-CoV (SPs, top right); it can be secreted by the cell (top right) or processed by proteasomes to yield antigenic fragments that are presented on MHC class I surface receptors (step 3). These teach cytotoxic T cells to recognize the antigen and trigger apoptosis in virus-infected cells (step 4). The SP molecules liberated to the extracellular space get phagocytosed in an autocrine fashion to feed MHC class II molecules via endo-lysosomes (step 5) to “educate” Th and B cells for specific neutralizing antibody production (step 6). Abbreviations: BCR, B cell receptor; ER, endoplasmic reticulum; TCR, T-cell receptor; BCR, B-cell receptor. Reproduced with permission from Ref. [40] 2025, Springer Nature.

**Figure 2 pharmaceutics-17-01327-f002:**
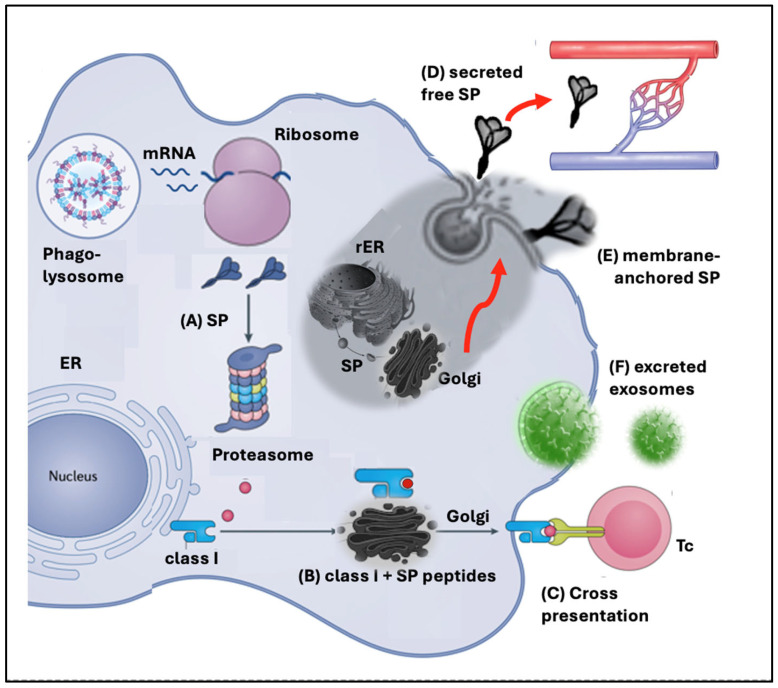
Different fates of the spike protein (SP) following synthesis on free cytoplasmic and ER-bound ribosomes. (**A**) SP synthesized on free cytoplasmic ribosomes is degraded by proteasomes into 8–12 amino acid (AA) peptides, which are transported into the endoplasmic reticulum (ER) via the TAP transporter, where they bind to MHC class I molecules. The peptide-class-I complexes then migrate through the Golgi apparatus to the plasma membrane (**B**) for cross-presentation to cytotoxic CD8^+^ T cells (**C**). In contrast, SP translated on ribosomes bound to the rough endoplasmic reticulum (rER; dark gray field, upper right) follows the classical secretory pathway through the Golgi apparatus into secretory vesicles, resulting either in the release of free protein into the circulation (**D**), or in the “crowning” of the cells surface with membrane-anchored SP, as observed in SARS-CoV-2 (**E**). Additionally, cytoplasmic SP may be packaged into exosomes (green area) and excreted into blood, contributing to systemic dissemination of both the mRNA and the SP (**F**).

**Figure 3 pharmaceutics-17-01327-f003:**
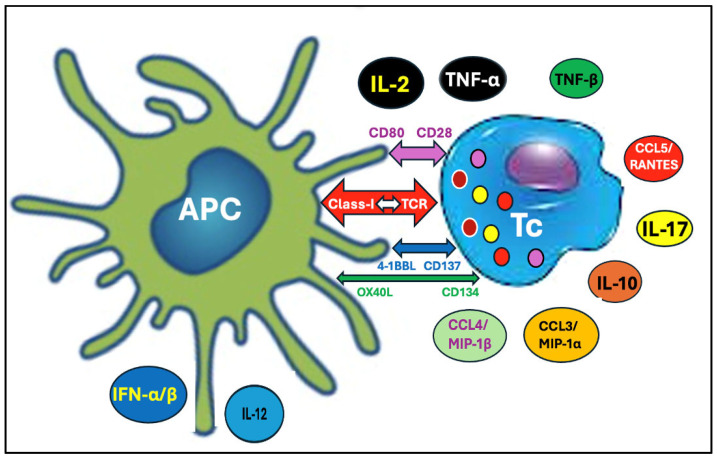
Synergistic immune cell activations involved in the potential autoimmune attack against APCs by SP-specific cytotoxic T cells under proinflammatory conditions. In the chain of autotoxicity of APCs, the first activation signal is the recognition of the SP peptide-MHC class I complex on APCs by the TCR of Tc. Next, the CD28 surface protein on Tc binds to B7-1 (CD80)/B7-2 (CD86) on APCs, and/or 4-1BB (CD137) on Tc to 4-1BBL on APCs, and/or OX40 (CD134) on Tc to OX40L on APCs. These interactions entail the activation of both cell types with the secretion of numerous proinflammatory cytokines and chemokines, some specified in the figure. It is hypothesized that the strong proinflammatory stimulus of mRNA-LNPs may augment or act synergistically with these natural immune interactions and thus induce autoimmune attack against the APC, ultimately apoptosis. The scheme applies to all body cells transfected with the vaccine and expressing SP fragments on their surface class I molecules. The double arrows illustrate ligand-receptor interactions, and the red and yellow-colored circles within the Tc represent the cytotoxic effector molecules released upon cytotoxic T cell attack.

**Figure 4 pharmaceutics-17-01327-f004:**
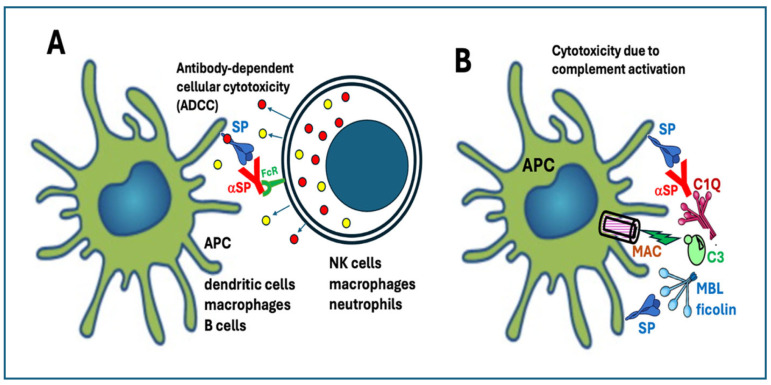
Antibody-mediated adverse impacts of mRNA vaccines after the rise in specific anti-SP antibodies. (**A**) Anti-SP specific antibody (aSP)-dependent cellular cytotoxicity (ADCC), whereupon natural killer (NK) cells, macrophages or neutrophils recognize the cell-bound antibodies’ Fc portion via their Fc receptor (FcR) and release cytotoxic granules (colored dots) inducing apoptosis. (**B**) Cell-bound antibodies initiate classical pathway C activation starting with the binding of C1q and ending in the formation of the membrane attack complex (MAC), which causes membrane damage. The SP on the APC surface also activates C via the lectin pathway, by binding mannose binding lectin (MBL) or ficolin. The result is the same: MAC-mediated cytotoxicity.

**Figure 5 pharmaceutics-17-01327-f005:**
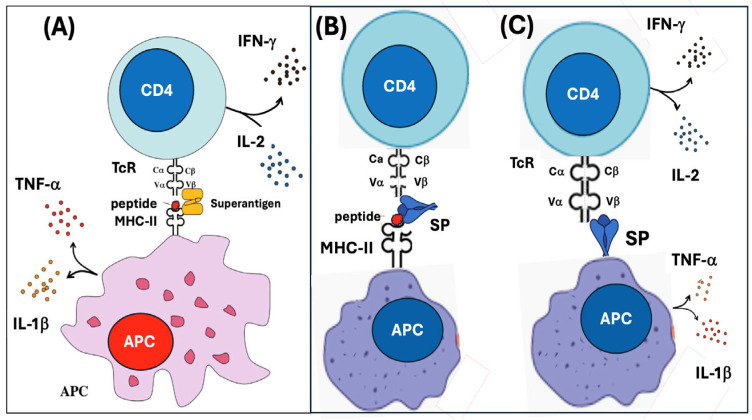
Illustration of the effects of superantigens. (**A**) A bacterial superantigen (e.g., *Staphylococcus Enterotoxin-B*) can crosslink the MHC class II receptor on APCs directly to the variable region of the T cell receptor (TCR) β-chain (TcR Vβ). This leads to massive, non-specific stimulation of both APCs and T cells, resulting in the release of multiple inflammatory cytokines. (**B**) Hypothetic crosslinking of the MHC class II receptor on APCs to the Vβ region of TcRs by the SP, forming an APC-SP-T cell ternary complex. This linkage may or may not stimulate cytokine production. (**C**) Hypothetical scheme of the mechanism of SP-induced APC and T cell activation whereupon the superantigenic motif on APC-exposed S1 directly binds Vβ on T cells. The subsequent activation steps involving furin are discussed in the text. (**A**) was adapted from Proft T, Fraser JD. Streptococcus pyogenes Superantigens: Biological properties and potential role in disease. 2022. In: Ferretti JJ, Stevens DL, Fischetti VA, editors. Streptococcus pyogenes: Basic Biology to Clinical Manifestations, 2nd edition, University of Oklahoma Health Sciences Center; 2022. https://www.ncbi.nlm.nih.gov/books/NBK587120/figure/superantigens.F1/, accessed on 30 September 2025. The CC-by BY-NC-ND 4.0 license allows free reuse of this figure.

**Figure 6 pharmaceutics-17-01327-f006:**
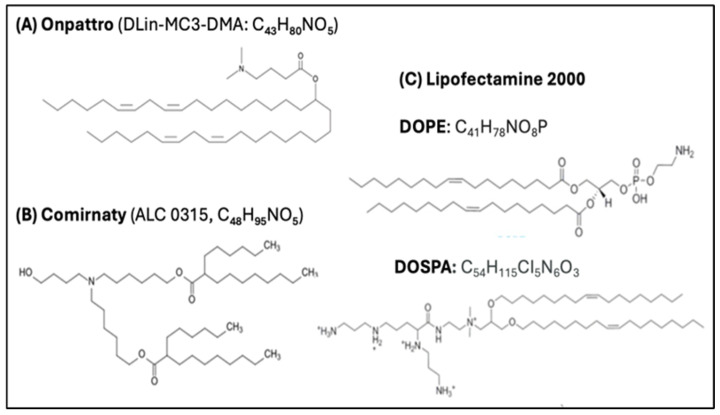
Chemical structures of ionizable lipids in (**A**) Onpattro (patisiran), (6Z,9Z,28Z,31Z)-heptatriaconta-6,9,28,31tetraen-19-yl-4-(dimethylamino) butanoate); (**B**) Comirnaty, ALC-0315, [(4-Hydroxybutyl)azanediyl]di(hexane-6,1-diyl) bis(2-hexyldecanoate; (**C**) Lipofectamine 2000, containing, DOPE (1,2-dioleoyl-sn-glycero-3-phosphoethanolamine) and DOSPA (2,3-dioleyloxy-N-[2-(sperminecarboxamido)ethyl]-N,N-dimethyl-1-propanaminium HCl salt) [22].

**Figure 7 pharmaceutics-17-01327-f007:**
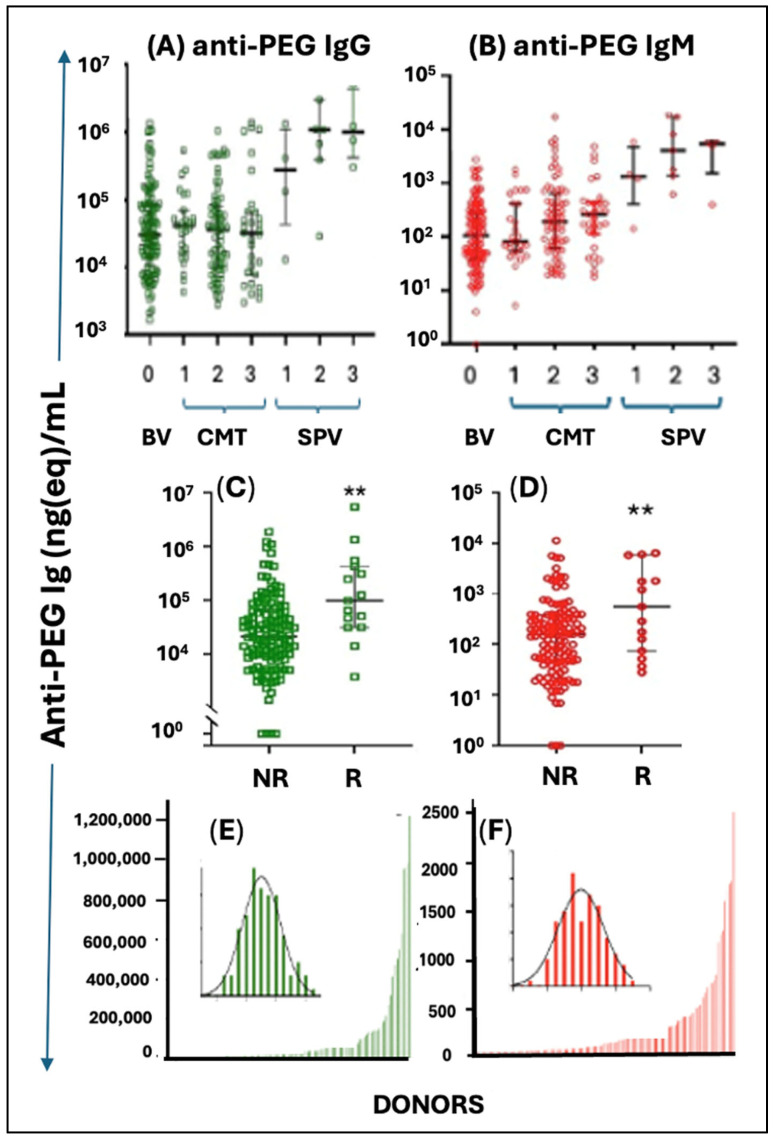
Anti-PEG antibody levels before and after mRNA vaccination and their correlation with anaphylactic reactions. (**A**) Anti-PEG IgG measured before vaccination (BV) and after the 1st, 2nd, and 3rd (booster) injections with Comirnaty (CMT) and Spikevax (SPV). (**B**) Same as in (**A**), for anti-PEG IgM. (**C**,**D**) Comparison of anti-PEG IgG (**C**) and IgM (**D**) levels between reactors (R) exhibiting hypersensitivity reactions (HSRs) and non-reactors (NR). The symptoms in R subjects were Grade 2–3. The time interval between antibody determinations and vaccine-induced reactions varied in different subjects within a 4-months window. The ** mean *p* < 0.01, respectively (*t*-test of log-transformed data) imply significantly increased risk for anaphylaxis in people with high anti-PEG antibody levels. (**E**,**F**) Pre-vaccination anti-PEG antibody concentrations in a mixed population of nearly 120 blood donors. Anti-PEG IgG (**E**, **green**) and anti-PEG IgM (**F**, **red**) values were arranged in ascending order. The insets show probability distribution of the log-transformed antibody data, which passed the Shapiro–Wilk normality test. Such log-normal distribution is consistent with the multicausal nature of anti-PEG antibody formation, where variability increases with the mean [229]. Reproduced with permission from Ref. [229] 2025, Elsevier.

**Figure 8 pharmaceutics-17-01327-f008:**
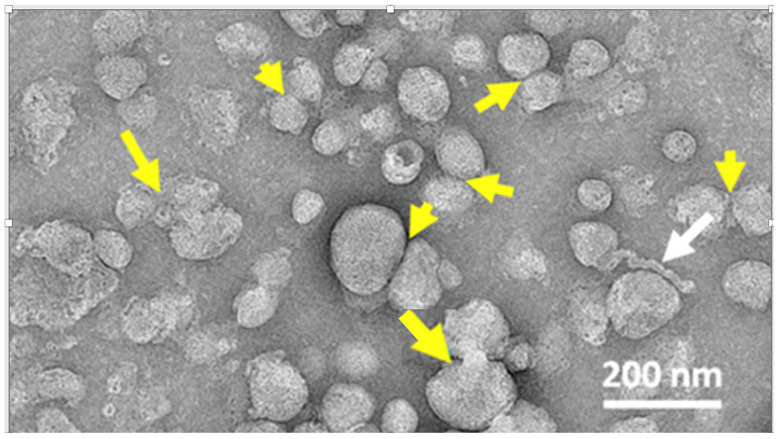
TEM images of a Comirnaty sample stored 7 days at 4 °C, diluted in water, and stained with acidic uranyl acetate as described in Ref. [15]. Yellow arrows point at the fusion of nanoparticles, and white arrows to elongated helical-like structures, possibly mRNA leaving an LNP. Modified from Ref. [15].

**Figure 9 pharmaceutics-17-01327-f009:**
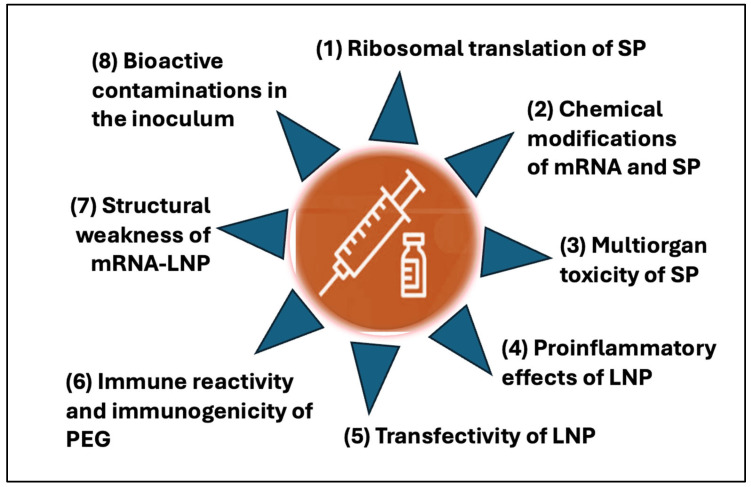
Inherent structural and functional features of mRNA vaccines that may contribute to AEs. The schematic illustrates eight interconnected and potentially synergistic vaccine characteristics that may underlie the unusually broad spectrum of AEs. These include (1) uncontrolled ribosomal translation of the SP, (2) chemical modifications of the mRNA and SP extending their untoward effects, (3) multiorgan toxicity of the SP, (4) innate immune reactivity of the LNP causing local or systemic inflammations, (5) the LNPs are proven transfection vectors, (6) the immune reactivity and immunogenicity of LNP-surface polyethylene glycol (PEG) causing CARPA or anaphylaxis, (7) the unstable (phospho)lipid coat makes the LNP prone to disruption, (8) DNA fragment and other undisclosed contaminants in the vaccine pose toxicity risks, including carcinogenicity. These features may, alone or in different combinations, underlie AE susceptibility.

**Table 1 pharmaceutics-17-01327-t001:** Physical and chemical properties of mRNA-LNP vaccines that can be associated with theoretical risks for AEs.

Unique Vaccine Properties	Collateral Immune Effects/ Unintended Processes	Clinical Manifestations
(1) Ribosomal translation may alter antigen fate and function	Uncontrollable cytoplasmic accumulation of SP Diversification of antigen processing and presentation Proteasome digestion Cross-presentation of SP peptides on MHC class I Increase in circulating CD8^+^ Tc cells Reduction in circulating memory T and CD4^+^ helper T cells SP secretion SP expression on cell surfaces Exosomal excretion of the SP Excessive somatic hypermutation in B cells IgG4 class switching Production of anti-idiotype antibodies Low IgG Fc glycosylation Frameshift mutation SP polymorphism Reverse transcription Insertion mutagenesis	New onset mono- or multiorgan inflammations Flare-up of existing inflammatory conditions Relapses of previous inflammatory diseases New onset mono- or multiorgan autoimmune conditions Flare-up of existing autoimmune conditions Relapses of previous autoimmune diseases Risk of persistent bacterial and viral infections Risk of recurring bacterial and viral infections Weakened immune response/immune suppression Acute/chronic clotting disorders Thrombosis Myocardial infarction Stroke Cytokine storm
(2) The SP has toxic effects	Complement activation Endothelial inflammation, microvascular damage Oxidative (mitochondrial) damage Cytokine release (Micro)thrombosis Red blood cell aggregation White cell activation and/or cytotoxicity Platelet activation and/or coagulation abnormalities Fibrin thrombus formation Blood–brain barrier damage	
(3) Multiple chemical modifications of the mRNA and the SP increase their activity and stability *	Uncontrollable cytoplasmic accumulation of SP Diversification of antigen processing and presentation Proteasome digestion Cross-presentation Spike protein secretion Spike protein expression on cell surfaces Exosomal excretion of the SP Excessive somatic hypermutation in B cells Frameshift mutation SP polymorphism Reverse transcription Insertion mutagenesis	Extension of all inflammatory and autoimmune symptoms listed for points (1) and (2) (Pseudo)allergies Anaphylaxis All symptoms of extended SP toxemia
(4) The LNP is a strong stimulant of innate immune responses, also enabling mRNA transfection	Germinal center hyper-reactivity Somatic hypermutation in B cells Enhancement of immunogenicity Activation of innate cellular immunity Persistent production of proinflammatory cytokines and chemokines Perpetual and intermittent activation of the complement system	
(5) PEG on the LNP surface is immune reactive and immunogenic	Complement activation Enhanced disintegration Anti-PEG and anti-LNP immunogenicity	
(6) The mRNA-LNP is unstable in aqueous media	Ready escape from the injection site into the lymph and blood Enhanced multiorgan distribution and transfection with mRNA lipoplexes Buildup of multimolecular assemblies	Pseudoallergic reactions (CARPA) Anaphylaxis Multiorgan inflammations Sensitization to PEG
(7) The spike protein is stabilized by enrichment with proline and guanine (G)-quadruplexes	Extended SP immunogenicity and toxicity	Complement activation Endothelial inflammation Microvascular damage Oxidative (mitochondrial) damage Cytokine release (Micro)thrombosis Red blood cell aggregation White cell activation and/or cytotoxicity Platelet activation and/or coagulation abnormalities Fibrin thrombus formation Blood–brain barrier damage
(8) The injectable vaccine may contain contaminations with plasmid DNA, double-stranded mRNA and/or inorganic elements or complexes	Reverse transcription Genomic integration of plasmid sequences Insertion mutagenesis with SV-40 promoter/enhancer sequences Accumulation of multimolecular assemblies Complement activation	Extended SP expression and toxicity Acute and/or chronic systemic inflammatory reactions Turbo cancer Hypersensitivity reactions

Abbreviations: SP, spike protein; CARPA, complement activation-related pseudoallergy; PEG, polyethylene glycol. * Same collateral immune effects/unintended processes as for ribosomal translation (point 1), except the changes are amplified and extended.

**Table 2 pharmaceutics-17-01327-t002:** Systemic changes underlying the pan-toxicity of the SP.

Systemic Toxicities	References
Complement activation	[147,148,149]
Endothelial inflammation, microvascular damage	[80,145,146,147,148]
Oxidative (mitochondrial) damage	[36,141,142,150]
Myocardium contractility decrease	[150]
Cardiac fibrosis	
Cytokine release	[143,144]
(Micro)thrombosis	[131,148,151,152,153]
Red blood cell aggregation	[84]
White cell activation and/or cytotoxicity	[93,154,155]
Platelet activation and/or coagulation abnormalities	[156,157]
Fibrin thrombus formation	[158]
Blood–brain barrier damage	[159]

## Data Availability

No new data were created or analyzed in this study. Data sharing is not applicable to this article.

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
