# Peer review of "Unique Features and Collateral Immune Effects of mRNA-LNP COVID-19 Vaccines: Plausible Mechanisms of Adverse Events and Complications"

_pharmaceutics, 2025, doi:10.3390/pharmaceutics17101327_

Round 1
Reviewer 1 Report
Comments and Suggestions for Authors
The author reviewed many aspects of immune responses induced by SARS-CoV-2 mRNA vaccine. He cited huge number of published studies, but there are a few issues.
- Lines 221-223, the author stated, “As a consequence, the SP peptides can trigger 221 cytotoxic T cell (Tc) attacks, one way to induce autoimmune phenomena that are 222 associated with mRNA-LNP vaccinations”. However, the author didn’t present robust studies supporting this sentence. In my opinion, this sentence should be the author’s hypothesis based on previous studies.
- Lines 255-258, the sentence “…, causing multiorgan toxicities” should be the author’s hypothesis.
- Line 364, please clearly point to which section is “Section 2.3” in the sentence “… be discussed later in Section 2.3 as an in dependent design problem …”.
- Lines 336-338, it is very confused for me to understand the sentence “Such virus-mimetic “pseudo-superinfection” may contribute to the AEs associated with mRNA vaccines”. The author should briefly explain the mechanism that exosomal excretion of SP results in AEs.
- Line 357, the author stated, “… as discussed in section 6.1”, but there is not section 6.1 in the manuscript. Please correct it.
- Lines 359-361, the author stated, “the autoimmune complications after mRNA vaccination may be explained with an overdrive of somatic hypermutation, at least in part”. Please explain the mechanism.
- Lines 576-578, the author stated, “GC enrichment, just as the uridine-y exchange, can also reduce the inflammatory response to the mRNA”. Please explain the mechanism.
- There are lots of typos, such as “at least in paert” and “Similar reactions and were reported”. Please correct them.
Reviewer 2 Report
Comments and Suggestions for Authors
I have the following comments to be considered
1- The authors should demonstrate in the abstract and the introduction the knowledge gap: what exactly is unknown in the field and how this study addresses it.
2- The authors should clarify the distinction between adverse effects caused by the mRNA vaccine platform itself versus those potentially attributable to vaccine adjuvants.
-
Currently, the manuscript does not make it clear whether reported complications are due to the active mRNA construct, the lipid nanoparticle (LNP) delivery system, or other adjuvant components.
-
Since the mechanism of immune activation and potential reactogenicity may differ between these elements, the authors should discuss this more explicitly in the Introduction and Discussion.
-
Relevant recent studies on LNP-induced innate immune activation and comparisons with traditional adjuvanted vaccines should be cited to strengthen this section.
3- The manuscript would benefit from a comparative discussion of whether mRNA vaccines from different suppliers (e.g., Pfizer-BioNTech vs. Moderna) or formulations with distinct adjuvant systems manifest similar or divergent side effects.
-
This is important because both the mRNA sequence/stability modifications and the lipid nanoparticle (LNP) composition can vary between manufacturers, which may influence immune activation and reactogenicity.
-
The authors should integrate available evidence from head-to-head observational studies or pharmacovigilance databases to highlight whether safety signals are consistent across platforms, or if certain formulations show unique adverse event profiles.
4- Figures should have clearer labels, legends, and resolution.
5- Plagiarism is too high
6- Name of organisms should be in italics throughout the manuscript
Round 2
Reviewer 1 Report
Comments and Suggestions for Authors
The author mentioned three references in the Response #1 to support the author's hypothesis. These references are about mechanisms of autoimmune response but not enough to conclude that SP protein cross-presentation could trigger autoimmune response. If the author want to confirm the hypothesis, he/she should perform a research or cite at least one previous study showing that intracellular expression of SP protein like mRNA vaccines indeedly results in antoimmune disease in vivo. The author have changed the statement to a hypothesis now so that I have no concern because I don't have any evidence to exclude this possibility.
Author Response
I thank the Reviewer for the thorough insight into the question of causality between spike protein (SP) cross-presentation and autoimmunity. As he/she acknowledged, we revised the text to indicate that such a cause-effect relationship is hypothetical. Indeed, we found no direct experimental evidence for such a relationship, but it is not easy, if possible, at all, to link an extremely complex autoimmune condition to a single factor, such as MHC-I cross-presentation of SP peptides. Such proof would require genetic studies analyzing MHC-I-SP peptide complexes in patients or animal models with autoimmunity, while excluding all other conditions as causes of autoimmunity. The complexity of such an experiment seems to exceed current clinical research capabilities, not to mention the lack of therapeutic or preventive implications other than avoiding cytoplasmic synthesis of the antigen. Thus, at present, the relationship remains hypothetical, based on several papers describing vaccination-induced cross-presentation of SARS-CoV-2 spike variants (e.g., Yin et al., 2023, Cell Reports, 42, 112470. https://doi.org/10.1016/j.celrep.2023.112470), the causal role of cross presentation in autoimmunity, and hundreds of papers reporting autoimmunity after mRNA vaccination. Overall, the Reviewer is correct, but fortunately he/she agrees to treat this mechanism as a hypothetical possibility is acceptable.
In response to the Reviewer's note regarding the English, I have made numerous changes in the text to improve clarity and grammar. Some professional corrections and updates have also been made as shown in the traced version. These do not alter the main messages just make them more accurate.

Reviewer 2 Report
Comments and Suggestions for Authors
None
Comments on the Quality of English LanguageThe English could be improved to more clearly express the research.
Author Response
We thank the reviewer for the time and feedback on the revised manuscript.